# Knocking out *TMEM38B* in human foetal osteoblasts hFOB 1.19 by CRISPR/Cas9: A model for recessive OI type XIV

**Laura Leoni**[1], **Francesca Tonelli**[1], **Roberta Besio**[1], **Roberta Gioia**[1],
**Francesco Moccia**[2], **Antonio Rossi**[1], **Antonella Forlino**[1]*

**1** Department of Molecular Medicine, Biochemistry Unit, University of Pavia, Pavia, Italy, **2** Department of Biology and Biotechnology "L. Spallanzani", Laboratory of General Physiology, University of Pavia, Pavia, Italy

* antonella.forlino@unipv.it

**Data Availability Statement:** All relevant data are within the manuscript and its Supporting Information files.

## Abstract

Osteogenesis imperfecta (OI) type XIV is a rare recessive bone disorder characterized by variable degree of severity associated to osteopenia. It is caused by mutations in *TMEM38B* encoding for the trimeric intracellular cation channel TRIC-B, specific for potassium and ubiquitously present in the endoplasmic reticulum (ER) membrane. OI type XIV molecular basis is largely unknown and, due to the rarity of the disease, the availability of patients' osteoblasts is challenging. Thus, CRISPR/Cas9 was used to knock out (KO) *TMEM38B* in the human Foetal Osteoblast hFOB 1.19 to obtain an OI type XIV model. CRISPR/Cas9 is a powerful technology to generate *in vitro* and *in vivo* models for heritable disorders. Its limited cost and ease of use make this technique widely applicable in most laboratories. Nevertheless, to fully take advantage of this approach, it is important to be aware of its strengths and limitations. Three gRNAs were used and several KO clones lacking the expression of TRIC-B were obtained. Few clones were validated as good models for the disease since they reproduce the altered ER calcium flux, collagen I structure and impaired secretion and osteoblastic markers expression detected in patients' cells. Impaired proliferation and mineralization in KO clones unveiled the relevance of TRIC-B in osteoblasts functionality.

## Introduction

In nature calcium represents the third most abundant ion and it has a fundamental role in modulating cell activity by acting as second messenger for several cellular biological processes, ranging from differentiation, proliferation, metabolism, autophagy to apoptosis [1]. To enable such multiple responsive behavior, the cytosolic calcium concentration is kept extremely low through sequestration into organelle stores [2]. This allows quick and transient calcium spikes upon intracellular release mediated by voltage, or ligand dependent, calcium-permeable channel openings [3–6]. It is the rapid change in cytosolic concentration, responsible for the activation of calcium dependent proteins and/or enzymes, that mediates cellular response to external signals [7]. The main intracellular calcium store is represented by the sarcoplasmic-/

**Funding:** This research was funded by a Grant of the Italian Ministry of Education, University and Research (MIUR) to the Department of Molecular Medicine of the University of Pavia under the initiative "Dipartimenti di Eccellenza (2018–2022)" and Telethon Exploratory Grant GEP15066 to A.F.. The funders had no role in study design, data collection and analysis, decision to publish, or preparation of the manuscript.

**Competing interests:** The authors have declared that no competing interests exist.

endoplasmic- reticulum (SR/ER) in excitable and non-excitable cells, respectively, and its release is guaranteed by ryanodine receptors (RyRs) and inositol 3 phosphate receptors (IP$_3$Rs), respectively [8]. The maintenance of SR and ER membrane potential following calcium release requires the opening of cation channels allowing the passage of counter ions from the cytosol to the SR/ER lumen [9]. The trimeric cation channels, TRICs, were identified as responsible for the transport of potassium ions across the SR/ER membranes mediating counter cations fluxes necessary for the calcium release [10]. The mammalian family of TRICs is composed by TRIC-A and TRIC-B subtypes, which are encoded by two different genes: *TMEM38A* and *TMEM38B* [11]. TRIC-A couples with RyRs and it is mainly expressed in muscle and brain, while TRIC-B synchronizes with IP$_3$Rs and is ubiquitously expressed. *Tric-a* knock out mice show impaired muscle contraction [11], whereas *Tric-b* knock out are lethal at birth due to insufficient surfactant production and alveoli collapse [12]. Surprisingly, *Tric-b* mutant mice revealed a bone defect associated to reduced extracellular matrix (ECM) and impaired mineralization, with no evident muscle or neurological phenotype [13]. In 2016, Cabral et al. found TRIC-B loss-of-function in individuals diagnosed with a recessive form of osteogenesis imperfecta (OI), classified as OI type XIV [14]. Individuals with mutation in *TMEM38B* are characterized by osteopenia, but not respiratory failure, and they show a great phenotypic variability with mild to severe bone deformities, none to frequent fractures, different degree of growth retardation and short stature [14, 15]. Other typical OI features such as dentinogenesis imperfecta, grey-blue sclerae, hearing loss and scoliosis have also extremely variable penetrance. Both murine model and patients' osteoblasts present the expected impaired calcium flux from ER to cytosol, but the link between TRIC-B mutations and bone phenotype is still poorly defined and its elucidation will require further investigation using appropriate tools.

*In vitro* cell cultures represent one of the most consolidate models to study cell behavior and to dissect biochemical pathways in normal or pathological condition, independently from the external contest, which can be the tissue, the organ, or the entire organism. Primary osteoblasts, the cells of choice to investigate brittle bone diseases, such as OI type XIV, are extremely difficult to obtain from patients due to the invasiveness of bone biopsy. Furthermore, osteoblasts need to be used at very early passages to avoid their *in vitro* de-differentiation to fibroblast-like cells [16]. To overcome osteoblast scarcity, heterogeneity and limited lifespan, the generation of immortalized cells lines represents the favorite tool in bone research, since they do not need isolation from fresh biopsy, are easy to maintain and are characterized by relative phenotypic stability [16]. Here we used CRISPR/Cas9 to successfully generate the first *in vitro* osteoblast model for OI type XIV reproducing the molecular and biochemical features of patients' cells and we demonstrated the negative effect of TRIC-B loss-of-function in osteoblast proliferation and mineralization. The technical limitation and strength of CRISPR/Cas approach in providing *in vitro* disease models is also discussed.

## Materials and methods

### Human foetal osteoblast (hFOB) cell line

The immortalized human foetal osteoblast hFOB 1.19 cell line (ATCC, CRL-11372) was used to generate the *TMEM38B* knock out model. Cells were grown at 34°C in humidified atmosphere containing 5% $CO_2$ in growing medium made of Dulbecco's Modified Eagle's Medium/Nutrient Mixture F-12 Ham (DMEM:F12) (Sigma Aldrich, St. Louis, Missouri, USA) containing 2.5 mM L-glutamine and 15 mM 2-[4-(2-hydroxyethyl)piperazin-1-yl] ethanesulfonic acid (HEPES) and added with 10% bovine serum (Euroclone, Buchs, Switzerland) and 0.3 mg/ml geneticin (G418).

For the differentiation and mineralization experiments, cells were kept in DMEM media (Lonza Biosciences, Basel, Switzerland) supplemented with 100 nM dexamethasone (Sigma Aldrich), 50 μg/ml ascorbic acid 2-phosphate (Fluka) and 10 mM β-glycerophosphate (Sigma Aldrich). Cells were maintained in this medium at 37˚ C for 4, 8, 15 or 21 days depending on the experiments.

For each experiment the not transfected human foetal osteoblasts were used as control.

## Knock out *in vitro* model generation

Details on CRISPR/Cas9 gene editing of *TMEM38B* in hFOB are reported in (S1 Materials and methods).

## Western blot

Cells were plated in 6 well/plate in DMEM:F12 media and grown at 34˚C to confluency. Following PBS wash, cells were lysed and sonicated in RIPA buffer (150 mM NaCl, 1% IGEPALR CA-630, 0.5% sodiumdeoxycholate, 0.1% SDS, and 50 mM Tris, pH 8) supplemented with protease inhibitors (13 mM benzamidine, 2 mM N-ethylmalemide, 5 mM ethylenediaminetet-raacetic acid, 1 mM phenylmethylsulfonyl fluoride and 2 mM $NaVO_3$). Proteins were quantified by QuantumProtein Bicinchoninic Protein Assay Kit (Euroclone). Bovine serum albumin (BSA) (Sigma Aldrich, Darmstadt, Germany) was used as standard. Proteins extracted from WT and mutant clones were run on 6%, 7.5%, 10% and 12% (v/v) SDS-PAGE for COL1A1, ITPR1/2/3, OSX and TRIC-B, respectively and the gels were electrotransferred to a PVDF membrane (GE Healthcare, Chicago, Illinois, USA) at 100 V for 2 h in 19 mM Tris-HCl, 192 mM glycine and 20% (v/v) methanol. Membranes were then blocked with 2.5% milk for COL1A1 and with 5% (w/v) milk for the other proteins detection in TBS-Tween (20 mM Tris-HCl, 500 mM NaCl, pH 7.5 (TBS), 0.05% (v/v) Tween-20) (Sigma Aldrich, Darmstadt, Germany) at RT for 1 h and incubated o/n at 4˚C with anti-COL1A1 (Sigma Aldrich, St. Louis, Missouri, USA) 1:500 in 2.5% milk, with antibody against ITPR1/2/3 isoforms (Santa Cruz Biotechnology, Dallas, Texas, USA) 1:200 in 5% milk, with antibody against OSX (Invitrogen, Rockford, Illinois, USA) 1:200 in 1.5% milk or with antibody against TRIC-B (Invitrogen, Rockford, Illinois, USA) 1:500 in 2.5% milk, in TBS-T.

The anti-rabbit (1:10000 in 2% milk in TBS-T) or anti-mouse (1:2000 in 5% milk in TBS-T) secondary antibody (Cell Signaling, Danvers, Massachusetts, USA) was added for 1 h at RT. Total protein staining (Swift Membrane stain, G-Biosciences) or β-actin were used for protein loading normalization.

The signal was detected by ECL western blotting detection reagents (GE Healthcare, Chicago, Illinois, USA) and images were acquired with Image Quant LAS 4000 (GE Healthcare). Bands intensities were evaluated by densitometry, using ImageQuant TL analysis software (GE Healthcare).

## [$Ca^{2+}$] measurements

The resting [$Ca^{2+}$]$_i$ and ER-dependent $Ca^{2+}$ release were evaluated in untransfected and mutant clones loaded with the $Ca^{2+}$-sensitive fluorophore, Fura-2/AM (Molecular Probes Europe BV, Leiden, The Netherlands). The $Ca^{2+}$ imaging set-up and loading procedures were extensively described elsewhere [17]. Briefly, $2\times10^5$ cells were plated on 9 mm diameter cover slip (Marienfeld GmbH, Lauda-Königshofen, Germany) in a 6 well plate. Cells were incubated with 4 μM FURA-2/AM in 150 mM NaCl, 6 mM KCl, 1.5 mM $CaCl_2$, 1 mM $MgCl_2$, 10 Mm glucose and 10 mM Hepes (PSS) for 20 min at 37˚C in the dark. Cells were then washed in PSS solution and analysed with a fluorescence microscope equipped with Zeiss Achroplan 40X

lens. Cells were excited at 340 and 380 nm and the light emitted was revealed at 510 nm. The calibration of resting intracellular $Ca^{2+}$ levels was done using Grynkiewicz, as shown in [18]. UTP (100 μM) was added for 5 min, in absence of external calcium (0 $Ca^{2+}$), to stimulate calcium release from ER through the $IP_3R$ channel.

## Collagen analysis

Labelling of collagen with L-[2,3,4,5-3H]-proline (PerkinElmer, Milan, Italy) was used to evaluate collagen overmodification in un-transfected, WT and mutant clones. $2.5 \times 10^4$ cells were plated and cultured in growing medium in 6-well plates for 24 h. Cells were then incubated for 2 h with serum-free growing medium supplemented with 100 μg/ml (+)-sodium L-ascorbate (Sigma Aldrich). Labelling with 28.57 μCi of $^3$H-Pro/ml was performed for 18 h in the same medium. Collagen was extracted by pepsin digestion and salt precipitation as described [19]. Collagen was resuspended in Laemmli buffer and the radioactivity [counts per minute (CPM)] was measured using a liquid scintillation analyzer (PerkinElmer TRI-CARB 2300 TR). Equal amount of $^3$H-labeled collagen from un-transfected, WT and KO clones was loaded on 6% urea-SDS gels in non-reducing condition. The gels were fixed in 45% methanol, 10% glacial acetic acid, incubated for 1 h with enhancer (PerkinElmer), washed in deionized water and dried. $^3$H gel radiographs were obtained by direct exposure of dried gels to hyperfilm (Amersham) at −80˚C. The radiography films were digitalized by VersaDoc 3000 (Bio-Rad). The faster migration of the collagen bands in mutant samples was determined qualitatively in comparison to control samples loaded in the same gel. To quantify the percentage of collagen secretion, the ratio between the CPM in the media and in media plus cell layer was calculated.

## Real time qPCR

RNA was extracted from cell layer after 0, 4, 8 and 21 days of differentiation using QIAzol (Qiagen, Hilden, Germany) following manufacturer's recommendation. RNA was also extracted after 1 day of differentiation with or without incubation with 50 μM 2-APB for 16 h. RNA concentration was evaluated by Nanodrop spectrophotometer (Cellbio ND-1000), and RNA quality was check by agarose gel electrophoresis. cDNA was synthesized from 500 ng of RNA using the High Capacity cDNA Reverse Transcription kit (Applied Biosystems, Waltham, Massachusetts, USA) according to manufacturer's protocol in a final volume of 20 μL. qPCR was performed in triplicate in a 25 μL reaction using Taqman Universal PCR Master mix (Applied Biosystems, Waltham, Massachusetts, USA) and commercial TaqMan probes: Hs01047973_m1, *RUNX2*; Hs01866874_s1, *OSTERIX/SP7*; Hs01587814_g1, *BGLAP*; Hs00164004_m1, *COL1A1*, Hs99999905_m1, *GAPDH* using a Mx3000P (Stratagene) thermocycler and the MxPro software (Stratagene). The relative expression level of each gene was calculated using the ΔΔCt method. qPCR for *ITPR1*, *ITPR2*, *ITPR3*, *ALP*, *OPN*, *IBSP*, *DMP*, *RANKL*, *OPG*, *G6PD* and *GAPDH* (primers sequences are available upon request) was performed in 25 μL reaction mixtures with 12.5 μL SYBR Green Master mix (Applied Biosystems) using the QuantStudio 3 thermocycler and the QuantStudio Design & analysis software (Applied Biosystems). The relative expression level of each gene was calculated using the ΔΔCt method.

## Alkaline phosphatase assay

Alkaline phosphatase activity was analyzed on un-transfected cells and KO clones at 0, 4, 8 and 15 days of differentiation using the alkaline phosphatase assay kit (Abnova, Taipei, Taiwan), following the manufacturer's recommendation. Briefly, $2 \times 10^5$ cells were plated in 33 mm petri dish in DMEM media. The day after, the differentiation factors (100 nM dexamethasone,

50 µg/ml ascorbic acid 2-phosphate and 10 mM beta-glycerophosphate) were added to the media and the cells were moved to the 37˚C incubator. Cells were then lysated with 0.2% Triton-X-100 and quantified with Quantum Protein kit (Euro- clone, Pero, Italy). The absorbance (405 nm) was measured using a plate reader Clario star (BMG Labtech, Ortenber, Germany) and the results were normalized to day 0.

## Extracellular matrix mineralization

$2\times10^4$ cells were plated in 24 well/plate in growing medium and cultured at 34˚C. Two days after the medium was changed with DMEM supplemented with differentiation factors (100 nM dexamethasone, 50 µg/ml ascorbic acid 2-phosphate and 10 mmol/L beta-glycerophosphate). The medium was changed every two days for 15 days. Then, cells were fixed in 10% (v/v) formaldehyde (Sigma, St. Louis, Missouri, USA) at RT for 30 min, washed with $dH_2O$ and stained with 58 mM alizarin red s (Sigma–Aldrich) pH 4.1–4.3 for 45 min at RT in the dark. Images were acquired using a DFC480 digital camera (Leica Microsystems srl, Milan, Italy) connected to a light microscope (Dialux 20, Leica Microsystems srl, Milan, Italy).

For quantification 200 µL of 10% (v/v) acetic acid were added to each well, and the plate was incubated at room temperature for 30 min with shaking. Cells were scraped and transferred to a 1.5-mL tube. After vortexing for 30 s, the sample was heated at 85˚C for 10 min, incubated on ice for 5 min, centrifuged at 20000 g for 15 min and 200 µL of the supernatant were moved in a new tube. Then 75 µL of 10% (v/v) ammonium hydroxide was added to neutralize the acid. Aliquots of the supernatant were read at 405 nm using the UVIDEC-340 double beam spectrophotometer (Jasco, Lecco, Italy).

## Cell proliferation

Cell proliferation was evaluated by Cell Titer 96 AQueous Cell proliferation Assay (Promega, Madison, Wisconsin, USA), containing MTS [3-(4,5-dimethylthiazol-2-yl)-5-(3-carboxy-methoxyphenyl)-2-(4-sulfophenyl)-2H-tetrazolium salt] following the manufacturer's indication. $5x10^3$ cells were plated in 96 well/plate in complete growing medium at 34˚C and 37˚C. After 1, 2, 3 and 4 days of culture the medium was replaced with 100 µL of fresh medium. After 1 hour the MTS reagent was added and incubated at 34˚C and 37˚C for 4 hours. The absorbance was measured at 490 nm with a plate reader Clario star (BMG labtech). The values were normalized to day 1 of proliferation. Also, after 4 days of culture, the medium was replaced with 100 µL of fresh medium with or without 50 µM 2-APB plus the MTS reagent and incubated at 37˚C for 4 hours and absorbance was measured at 490 nm. Technical triplicates were performed.

## Statistical analysis

All values were expressed as mean ± standard deviation (SD) or standard error mean (SEM), as indicated. Technical triplicates were performed for all experiments, except for ALP assay for which technical duplicates were performed. Statistical comparisons were based on Student's T-test considering statistically significant p value $< 0.05$.

## Results

### CRISPR/Cas9 mediated TMEM38B knock out in hFOB

The most severely compromised cells in OI type XIV, whose dysfunction is causing the osteopenic phenotype, are osteoblasts [14, 15], thus to investigate the molecular basis of the disease, the generation of an *in vitro* model is recommended. To this purpose, the immortalized

human foetal osteoblasts line hFOB 1.19 was selected. These cells have the unique property to behave as immature- and mature- osteoblasts depending on growth conditions [20]. To knock out *TMEM38B* in hFOB 1.19, the CRISPR/Cas9 was chosen as gene editing technique, due to its ease of use and rapidity (**Fig 1A, S1 Results**).

Since knock out is more efficiently obtained by targeting genes at the 5' end, three RNA guides (gRNAs) were selected in this region using two freely available bioinformatic tools (**S1 Materials and methods, S1** and **S2 Figs**). Sixtythree single clones were obtained by serial dilutions of cells transfected with gRNA-2 (N = 17) and gRNA-3.2 (N = 36), separately or with a mix of the two guides (N = 10) (**S1–S3 Tables**). The absence of TRIC-B in the knock out (KO) clones was demonstrated by western blotting (**Fig 1B, S3A Fig**) and Sanger sequencing was carried out to determine the specific mutations on selected clones, 7 obtained from gRNA-2, 5 from gRNA-3.2 and 2 from the mix of the two. Small insertions and/or deletions (*indels*), responsible either for premature stop codon formation or change in amino acid sequence were identified. Of note, in 9 out of 14 sequenced clones more than two different mutant alleles were identified (**Table 1**), as expected based on 43% diploidy and 57% tetraploidy reported for hFOB 1.19. Even if the lack of full length TRIC-B was demonstrated, the heterogeneity of the *indels* does not allow to exclude that in case of in-frame transcripts, they could be translated, although quickly degraded, and thus, impair cell homeostasis. Indeed, a similar possibility cannot be excluded for some of the mutant *TMEM38B* alleles described so far in OI patients [14, 15, 21–26].

## Electrophysiological and biochemical validation of knock out clones as model for the disease

The impairment of $K^+$ flux through mutant TRIC-B channel was reported to compromise the $Ca^{2+}$ release from the ER to the cytosol through $IP_3Rs$ in fibroblasts and osteoblasts isolated from OI type XIV patients and osteoblasts obtained from the newborn *Tmem38b* murine knock out model [13, 14]. Thus, ER $Ca^{2+}$ mobilization was evaluated in some hFOB KO clones.

The $Ca^{2+}$-sensitive fluorophore FURA-2 was used to analyse $Ca^{2+}$ concentration in both un-transfected hFOB and TRIC-B KO osteoblasts. Resting steady state intracellular $Ca^{2+}$ concentration ($[Ca^{2+}]_i$) was significantly decreased in all KO clones with respect to control (**Fig 2A**). The overall ER ability to release $Ca^{2+}$ in TRIC-B-deficient cells was then evaluated by exposing the cells to cyclopiazonic acid (CPA, 20 μM), a selective inhibitor of sarco/ endoplasmic reticulum $Ca^{2+}$-ATPase (SERCA) activity. CPA induced a transient increase in $[Ca^{2+}]_i$ due to passive ER $Ca^{2+}$ release and reflected ER $Ca^{2+}$ levels. CPA-induced ER $Ca^{2+}$ mobilization was not affected by the absence of TRIC-B (**S4A–S4C Fig**). Next, we assessed the impact of *TMEM38B* KO on physiological ER $Ca^{2+}$ release through $IP_3Rs$. When the $IP_3Rs$-mediated ER $Ca^{2+}$ release was stimulated by the $IP_3$-producing agonist UTP, $Ca^{2+}$ exit from the ER was significantly reduced in TRIC-B KO clones compared to controls, proving the impairment of $Ca^{2+}$ flux in mutant clones, as previously described both in human and mice cells lacking TRIC-B (**Fig 2B and 2C**). Since in non-excitable cells the intracellular calcium flux from the ER to the cytosol is dependent on the activity of the $IP_3Rs$ channels, encoded by *ITPR1, ITPR2, ITPR3*, respectively, their expression was analyzed by qPCR. A significant reduction was found for IP3R-1 and -3 (**Fig 2D**), but western blot analysis revealed no difference in protein expression (**Fig 2E, S3B Fig**) (A3 KO clone 1.12 fold differences compared to controls).

To validate our KO clones as appropriate *in vitro* model for OI type XIV, collagen was also investigated since under post-translational modification of the protein was reported by Cabral

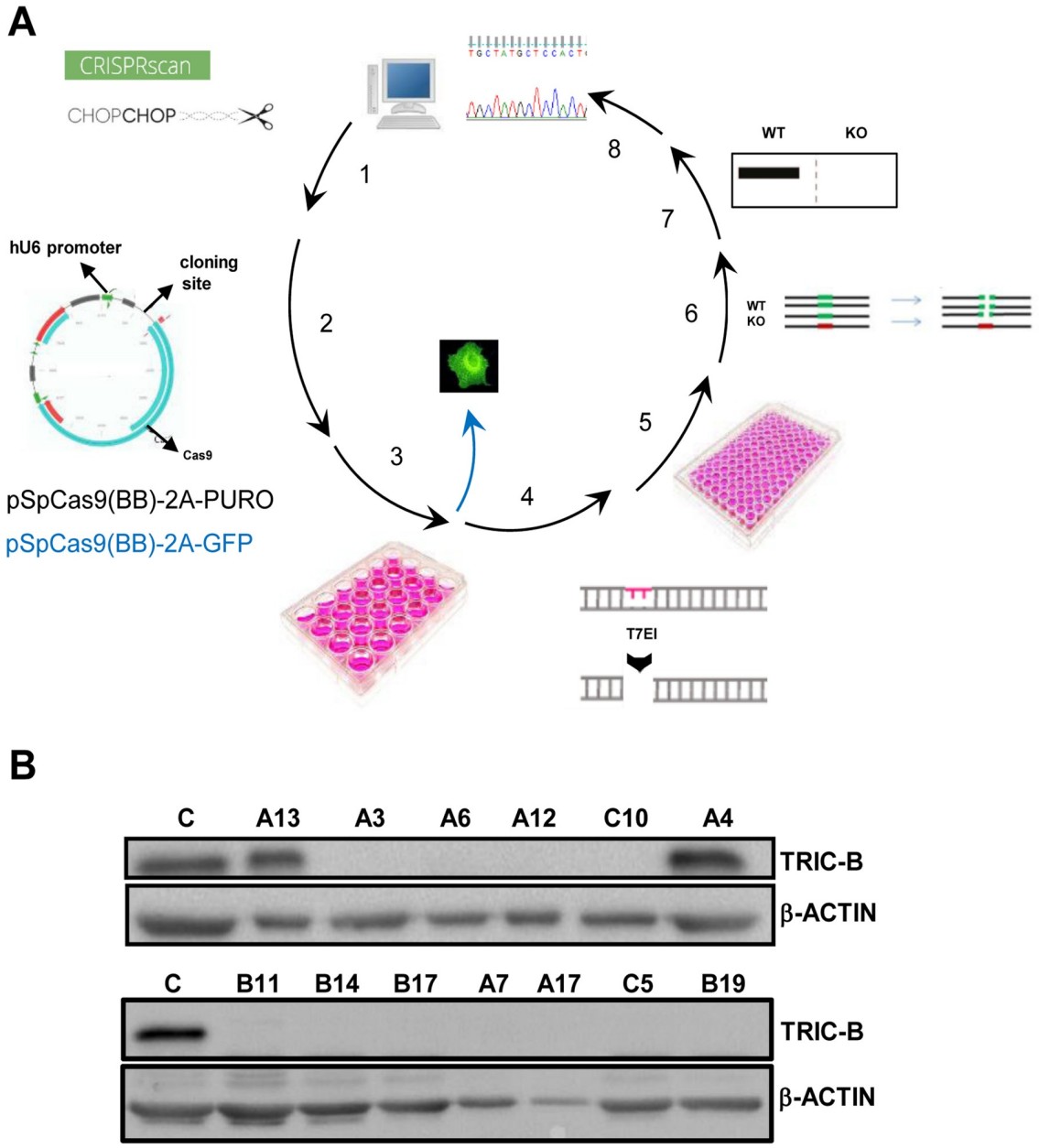

**Fig 1. OI type XIV *in vitro* model generation and western blot validation.** (A) Strategy used to generate *in vitro TMEM38B* models by CRISPR/Cas9 genome editing. (1) *In silico* target sequence identification; (2) cloning of gRNAs in the eukaryotic expressing vectors pSpCas9(BB)-2A-GFP and pSpCas9-2A-PURO; (3) hFOB cells lipofectamine transfection; (4) evaluation of targeting specificity by T7 endonuclease I assay and targeting efficiency by GFP screening (inside blue arrow); (5) generation of single cell clonal lines by serial dilutions of the cells transfected with the pSpCas9-2A-PURO constructs; (6) clonal line screening by restriction endonuclease digestion; (7) evaluation of TRIC-B protein by western blotting; (8) clones genotyping by sequencing. (B) Western blot analyses using TRIC-B specific antibody. The absence of protein expression is evident in mutant clones obtained from transfection with gRNA-2 (A3, A6, A7, A12, A17), gRNA-3.2 (B11, B14, B17, B19) and from transfection with both guides (C5, C10). Clones identified as WT (A4) or heterozygous (A13) from restriction enzyme digestion analysis expressed the protein similarly to not-transfected control cells (C).

et al in human patients' cells [14]. ³H-proline labelled collagen type I from TRIC-B KO clones revealed a faster migration of α(I) bands respect to control supporting reduced hydroxylation and glycosylation (**Fig 2F**).

**Table 1. Mutations identified in gRNA-2 and gRNA-3.2 transfected clones.**

| Clone | Guide | DNA mutations | Protein mutations |
|---|---|---|---|
| A3 | gRNA-2 | c.176_177ins132 | p.F59L; p.F59_G60ins44 |
| | | c.173_176delGTTT | *p.C58Lfs*72* |
| | | c.174delT | *p.F59Lfs*73* |
| A5 | gRNA-2 | c.176_177ins140 | *p.G60Lfs*63* |
| | | c.177_182delTGGTGG | p.F59_G60del; p.G61L |
| | | c.174delT | p.F59Lfs*73 |
| A6 | gRNA-2 | c.171_181del11ins636 | *p.H57Efs*113* |
| | | c.174delT | *p.F59Lfs*73* |
| | | c.171_187del17 | *p.C58Ffs*75* |
| A7 | gRNA-2 | c.176_177insG | *p.F59Lfs*81* |
| | | c.174delT | *p.F59Lfs*73* |
| A11 | gRNA-2 | c.173_174insT | *p.G60Wfs*81* |
| | | c.174delT | *p.F59Lfs*73* |
| A12 | gRNA-2 | c.176_177ins219 | *p.F59Lfs*61* |
| | | c.180_196del17insG>A | *p.G61Sfs*75* |
| | | c.174_175delTT | *p.F59Wfs*80* |
| A17 | gRNA-2 | c.177_178ins228 | p.F59_G60ins76 |
| | | c.177_182delTGGTGG | p.F59_G60del; p.G61L |
| | | c.173_188del16 | *p.F59Yfs*68* |
| B11 | gRNA-3.2 | c.414_454del41 | *p.N139Cfs*147* |
| | | c.399_449del51 | p.N134_A159del |
| | | c.426_454del29 | *p.I142Mfs*151* |
| B14 | gRNA-3.2 | c.435_437delAGC | p.A146del |
| | | c.426_437del12 | p.V143_A144del |
| B17 | gRNA-3.2 | c.431_434delTGAT | *p.M144Kfs*156* |
| | | c.427_447del21insG>A | p.V143_W149del; p.A150T |
| | | c.435_441del7 | *p.I145Mfs*155* |
| B19 | gRNA-3.2 | c.432_438del7 | *p.M144Ifs*155* |
| | | c.438_443delTATTGG | p.I147_G148del |
| | | c.433_437delATAGC | *p.I145Yfs*159* |
| | | c.437_438insT | *p.I147Yfs*162* |
| B24 | gRNA-3.2 | c.436_439delGCTA | *p.A146Lfs*156* |
| | | c.428_439del12 | p.M144_I147del |
| C5 | gRNA-2 | c.172_182delTGGTGG | p.F59_G60del;p.G61L |
| | | c.177delT | *p.F59Lfs*73* |
| | | c.159_209del51 | p.A54_A70del |
| | gRNA-3.2 | c.408_444del38 | *p.Y136Lfs*148* |
| | | c.437_443del7 | *p.A146Dfs*155* |
| | | c.436_446del9 | p.A146_G148del |
| | | c.429_438del10 | *p.M144Lfs*154* |
| | | c.432_437delGATAGC | p.M144_I145del; p.A146I |
| C10 | gRNA-2 | c.177delT | *p.F59Lfs*73* |
| | gRNA-3.2 | c.439_440delATins3549 | *p.I147Afs*156* |
| | | c.438_443delTATTGG | p.I147_G148del |

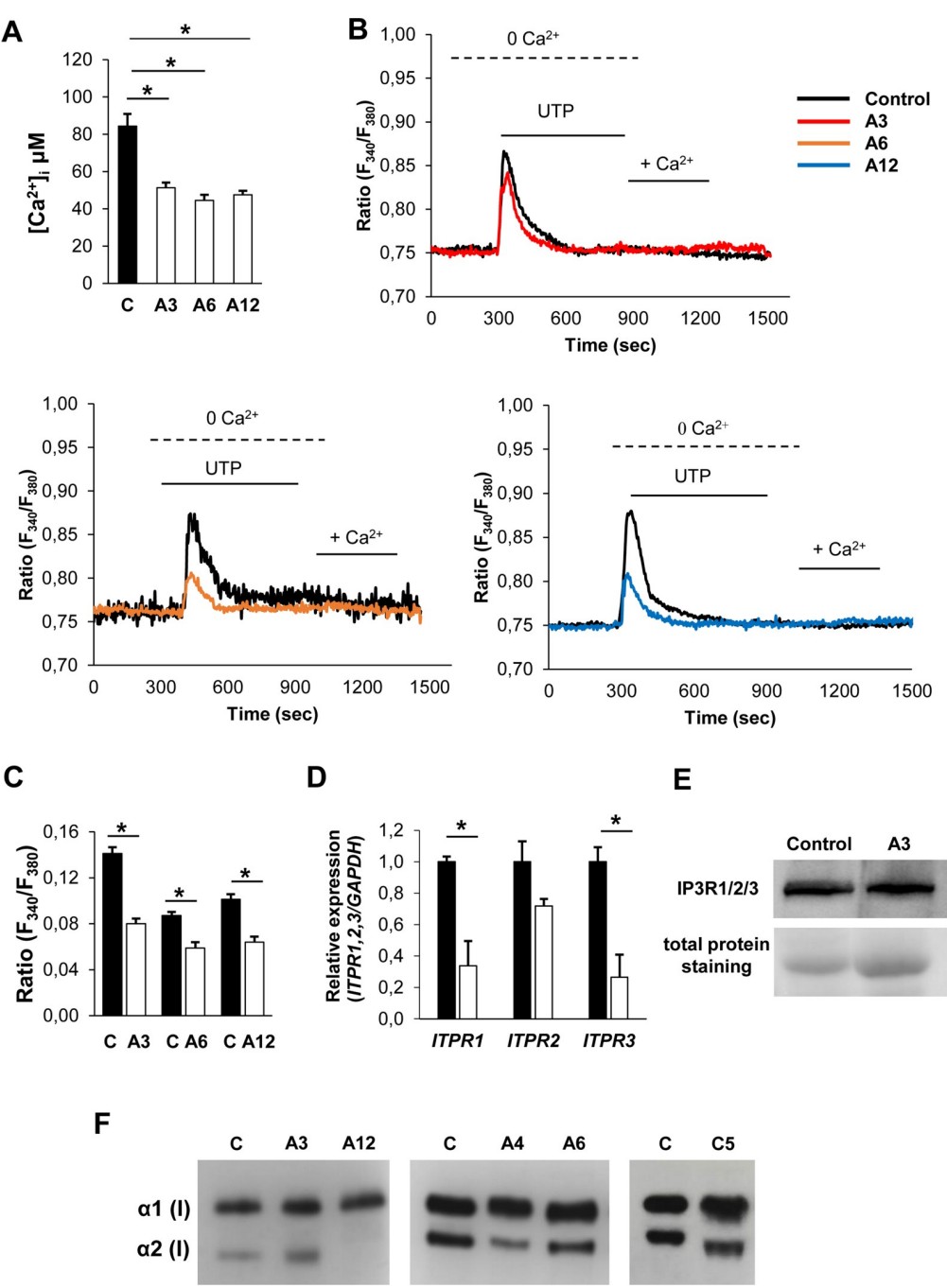

**Fig 2. Intracellular Ca²⁺ concentration, ER calcium flux and collagen analysis.** (A) Resting $[Ca^{2+}]i$ was reduced in KO clones (A3, A6 and A12) with respect to control cells. (B) Decreased $Ca^{2+}$ mobilization in TRIC-B deficient fibroblasts. UTP was used to stimulate the IP₃R channel in the absence of extracellular $Ca^{2+}$ (0 Ca²⁺). The calcium flux through IP₃R was significantly reduced in mutant (A3, A6 and A12) compared to control cells. The quantitative analysis is expressed as mean ± SEM in (C). (D) The expression of the *ITPR1*, *ITPR2*, *ITPR3* genes, encoding for the three isoforms of the IP₃Rs channels was evaluated by qPCR revealing a significant reduction for *ITPR1* and *ITPR3* in the KO clone A3. (E) Western blot analyses using IP₃R1/2/3 specific antibody revealed no differences in the channels protein expression. (F) Representative SDS-PAGE autoradiography of ³H-collagen type I extracted from control hFOB untrasfected cells (C), KO (A3, A12, A6 and C5) and WT (A4) clones. Collagen type I α bands show a faster electrophoretic migration in KO clones compared to untrasfected control and WT clone. *p<0.05.

### A3 KO clone proved to reproduce patients' osteoblast phenotype

Reduced expression of early osteoblastic markers was previously described in patients' osteoblasts, as well as in bone extract and osteoblasts from the Tric-b KO murine model [13, 15]. Thus, their expression was analyzed in few KO clones at day 0, 4 and 8 of culture in osteoblastic differentiation medium. Representative data obtained for A3 KO clone are shown from now on. The expression of *RUNX2*, *SP7*, *BGLAP* and *COL1A1* was significantly reduced in the mutant clone compared to controls (**Fig 3A**, **S5A Fig**). Similarly, a reduced expression of the early osteoblast markers was detected in control cells in which $Ca^{2+}$ flux was inhibited by 2-APB, supporting the calcium dependence of osteoblast differentiation (**S6A Fig**). The expression of the late osteoblastic markers *ALP*, *BGLAP*, *OPN*, *IBSP*, *DMP* was also evaluated following 21 days in osteoblastic differentiation medium (**Fig 3B**). *ALP* expression was reduced in A3 compared to controls. *BGLAP*, *IBSP*, and *DMP* expression was increased, as reported for patient' cells [15]. *OPN* was decreased, in agreement with the generally opposite roles of *IBSP* and *OPN* in promoting and inhibiting mineralization. Importantly, the decrease of collagen type I and osterix was also confirmed at protein level (**Fig 3C** **and** **S3C**, **S3D Fig**).

An increase of RANKL/OPG ratio in the mutant sample (**Fig 3D**) suggested an alteration in osteoclastogenesis possibly due to an osteoblast dependent mechanism.

### Impairment of TMEM38B KO cells proliferation and mineralization

Cell cycle progression is highly dependent on IP3-mediated ER $Ca^{2+}$ release [27, 28] thus, the effect on cell proliferation was evaluated in the KO clones. MTT analysis was performed both at 34°C, the temperature required for optimal hFOB proliferation as immature-osteoblastic cells, and at 37°C, the temperature at which human osteoblasts' grow.

A significant proliferation reduction was detected at all examined time points at both tested temperatures (**Fig 4A**, **S5B Fig**). The incubation of control cells with 2-APB, an inhibitor of ER calcium flux, mimicked the data of mutant cells, thus supporting the dependence of osteoblasts proliferation on $Ca^{2+}$ flux (**S6B Fig**).

To evaluate the effect of lack of TRIC-B on collagen production, the ratio of medium secreted collagen on total collagen synthesis was evaluated upon $^3$H-proline labelling of TRIC-B A3 KO clone. A significant reduction of collagen secreted into the medium was detected in mutant compared to control cells (**Fig 4B**). The mineralization was evaluated by measuring the ALP activity in cell lysates at 0, 4, 8 and 15 days of culture in osteoblastic differentiation medium. The KO clones showed reduced ALP activity with respect to control, supporting an impairment in mineralization (**Fig 4C**, **S5C Fig**). The delay in matrix mineralization was further confirmed in KO clones by quantitation of alizarin red S staining, following 15 days of *in vitro* culture in differentiation medium. The KO cells revealed a significant reduction in mineral deposition with respect to controls, even if the amount of mineral increased during differentiation in both control and mutants (**Fig 4D**, **S5D Fig**).

## Discussion

CRISPR/Cas9 was used to generate clonal lines lacking TRIC-B to develop an *in vitro* model for the recessive osteogenesis imperfecta form type XIV. TRIC-B was initially identified as a membrane protein participating in cellular $Ca^{2+}$ signaling and, based on homology with TRIC-A, was recognized as ER $K^+$ channel present in most of the tissues [11]. The characterization of the Tric-b knock out mouse and the identification of loss-of-function mutations in patients affected by OI type XIV demonstrated its unexpected and relevant role in bone homeostasis. Mutant murine and patients' cells are characterized by impaired calcium flux from ER, supporting the essential role of TRIC-B to mediate the counter cationic flux

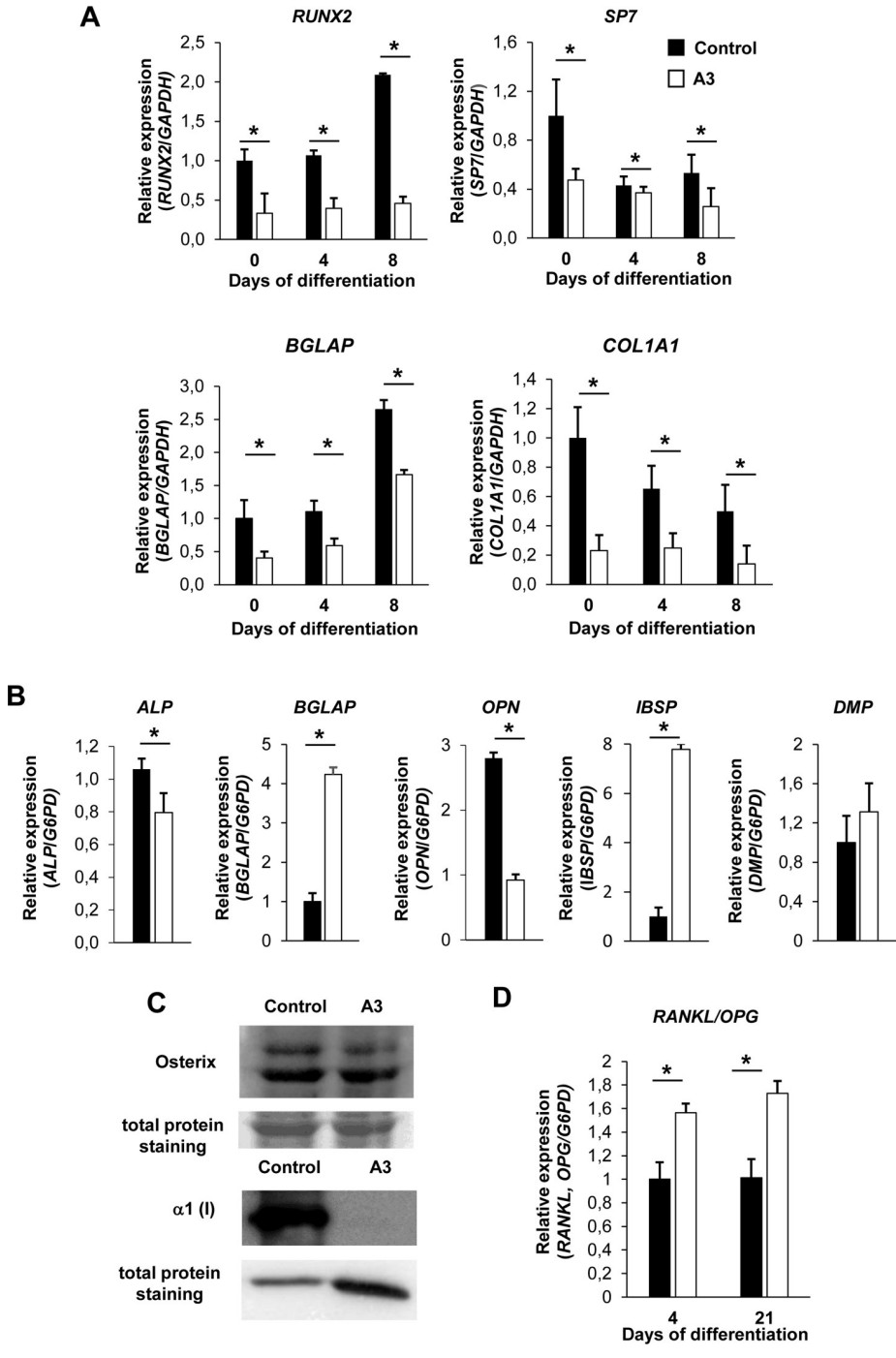

**Fig 3. Expression analysis of osteoblastogenic markers and of the RANKL/OPG ratio.** (A) *RUNX2*, *SP7*, *BGLAP* and *COL1A1* expression was reduced in A3 mutant clone with respect to control cells at all time points analysed during hFOB differentiation. (B) *ALP*, *BGLAP*, *OPN*, *IBSP*, *DMP* expression was evaluated following 21 days in osteoblastic differentiation medium. *ALP* and *OPN* expression was reduced while *BGLAP*, *IBSP* and *DMP* expression was increased in A3 mutant clone with respect to control cells. (C) A reduced collagen I and a slightly reduced osterix expression were detected by western blot analyses in mutant clone. (D) Measurements of RANKL/OPG ratio. qPCR analysis showed an increased ratio in the mutant clone A3 at both 4 and 21 days. * p<0.05.

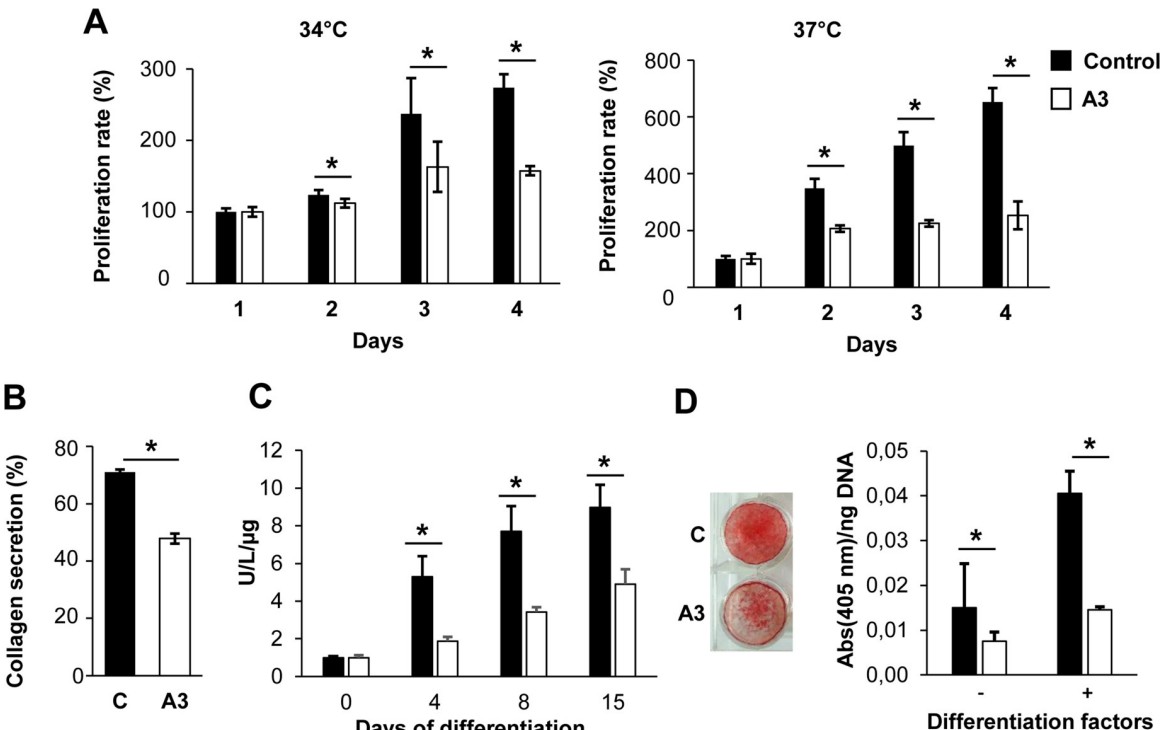

**Fig 4. Cell proliferation, collagen secretion and matrix mineralization analysis.** (A) Analysis of cell proliferation was performed both at 34˚C and 37˚C. A significant impairment of proliferation was detected in the KO clone A3 at both temperatures after 2, 3 and 4 days of culture. (B) Collagen secretion was evaluated upon cells ³H-proline labelling. A significant reduction of collagen secretion was detected in mutant clone compared to control cells. (C) The activity of ALP was significantly reduced in A3 mutant clone compared to WT at 4, 8 and 15 days of differentiation. All data were normalized to day 0. (D) The mineralization was analysed by alizarin red S staining. The mineral amount was significantly reduced in KO clones with respect to WT. *p<0.05.

necessary to limit the ER membrane hyperpolarization [13, 14]. Nonetheless, how altered calcium flux translates into a phenotype limited to bone is still puzzling the field. Of note, Tric-b KO mice die in perinatal period due to the insufficient surfactant release in the alveoli, whereas no respiratory phenotype was described in OI type XIV patients, suggesting different compensation in human versus murine pneumocytes, and anyway limiting the use of the mouse KO model to investigate the effect of TRIC-B through development [12, 13]. Cabral et al. reported in OI type XIV patients' cells an alteration in the expression of several calcium dependent proteins and enzymes involved in collagen folding and post translational modification including PDI, known to have a role in α(I) chain recognition as well as in proline hydroxylation, and LH1 and LH2 relevant for helical and telopeptide lysine hydroxylation. Furthermore, the structurally abnormal collagen was mainly intracellular retained and anyway not incorporated in the extracellular matrix, suggesting ER stress and matrix insufficiency as major players in the OI outcome [14]. The limited number of osteoblasts patients' cells analyzed associated to the variability in patients' outcome underlines the limit of using primary osteoblasts cells and the need for the newly generated *TMEM38B* knock out hFOB. The validation of the mutant clones was undertaken based on the phenotype previously described in patients' cells and compared to the findings reported in *Tmem38b* knock out mice. In detail, ER Ca²⁺ mobilization, early and late osteoblastic markers expression and collagen structure were analyzed. Resting steady state intracellular Ca²⁺ concentration ([Ca2+]i) was significantly decreased in all KO clones as previously detected in human cells and opposite to what found in mutant murine

osteoblasts [13, 14]. $Ca^{2+}$ exit from the ER was significantly reduced in TRIC-B KO clones compared to controls, confirming the impairment of $Ca^{2+}$ flux previously described both in human and mice cells lacking TRIC-B. Most of TRIC-B KO clones revealed faster electrophoretic α(I) bands reproducing the faster electrophoretic migration of collagen I from patients' osteoblasts and fibroblasts described by Cabral et al. and supporting a reduced level of post translational modification, associated with a decreased helical lysine hydroxylation [14]. One of the selected clones lacked the α2(I) band, likely due to an off-target effect of CRISPR/Cas9 targeting. Indeed, the risk of undesired mutations, other than the intended on-target site, is still one of the major drawbacks of CRISPR/Cas9 gene editing system, especially when in some selected sequences the guide choice is limited [29]. For the two effective guides chosen at the 5' end of the *TMEM38B* gene, gRNA-2 and gRNA-3.2, the CrisprScan predicted off-target sites with a maximum of 2 mismatches were one and none, respectively. The expression of early osteoblast markers (*RUNX2*, *SP7*, *COL1A1*) was significantly impaired in the mutant clone, reproducing both human and murine mutant osteoblasts findings [13, 15]. *ALPL* expression was reduced in A3 compared to controls, in agreement with the ALP activity data. The late markers *IBSP*, *BGLAP* and *DMP* were increased at day 21 of culture, as in the patient' cells. *OPN* was instead decreased, in agreement with the generally opposite roles of *IBSP* and *OPN* in promoting and counteracting mineralization [30]. Importantly, the increased RANKL/OPG ratio in the mutant sample at day 4 and 21 of culture suggested that the alteration in osteoclastogenesis detected in patients could be due to an osteoblasts' dependent mechanism as well as to an intrinsic defect. To complete the validation of *TMEM38B* KO *in vitro* model, the expression of *IP3R* was investigated. In patients' osteoblasts, but not in fibroblasts, a decreased expression was reported for *IP3R-1* to 16 ± 5% of control value without reduction of the protein level [14] and in TRIC-B KO clone similar findings were detected. In murine KO calvarial osteoblasts no RNA expression difference was shown for any of the isoforms [13]. Interestingly, among all clones generated only few, among which A3 was chosen as representative throughout the manuscript, pass through the validation analysis, thus underlining the relevance to have human mutant cells as comparison for the clones modified by CRISPR/Cas9 to reduce the number of clones to be screened. To know more on the role of TRIC-B in bone cells activity, proliferation, collagen secretion and extracellular matrix mineralization were investigated using the mutant hFOB OI type XIV model. The significant proliferation reduction detected in TRIC-B A3 KO clone, together with the low expression level of the early and higher expression of late osteoblastic markers, supports the low number of osteoblasts found in the patients' bone biopsies and in the diaphysis of Tric-b KO mouse long bones [13, 15]. Of note, no difference in proliferation was detected in mutant calvarial osteoblasts from the KO mouse model [13]. The reduced collagen secretion and ALP activity in TRIC-B clone explains the reduced collagen deposition and impaired mineralization found in both membranous and endochondral bones of newborn Tric-b KO mice associated to intracellular collagen retention in the ER and the reduced collagen and mineral amount in the extracellular matrix of long term cultured mutant murine osteoblasts [13]. Altered mineralization was also described in patients' bone biopsies [15]. Indeed, collagen I represents the most abundant protein scaffold for mineral deposition in the extracellular matrix and requires the activity of tissue non-specific alkaline phosphatase (ALP) that hydrolyzes pyrophosphate providing inorganic phosphate to promote mineralization. Interestingly, reduced expression of *Alp* transcript was reported in mutant murine osteoblasts, contrarily to its increase observed in patients' cells in which anyway protein activity was not evaluated.

## Conclusions

The *TMEM38B* KO hFOBs represent the first immortalized human osteoblast model for OI type XIV and they will represent a unique tool to address the still open questions on OI disease pathophysiology: how calcium flux impairment is affecting collagen synthesis and structure; which calcium dependent signaling pathways are mostly affecting osteoblast differentiation and activity; is calcium necessary for proper collagen and/or other relevant bone proteins secretion?

The mutant cells reproduced the reduced steady state $[Ca^{2+}]_i$, the impairment of IP$_3$Rs-mediated ER $Ca^{2+}$ release, the reduced expression of osteoblastic markers and the reduced *ITPR1* transcript found in patients osteoblasts. The impairment in cell proliferation of KO clone explains the reduced number of osteoblasts in patients' biopsies. Interestingly, in hFOB a decrease in alkaline phosphatase activity and a strong reduction in mineral deposition support the osteopenia and low bone volume described as common features in OI patients. The availability of an *in vitro* model will allow to dissect the molecular basis of the disease offering a valid platform for drug target discovery and preclinical screening.

## Supporting information

**S1 Materials and methods.**
(DOCX)

**S1 Results.**
(DOCX)

**S1 Fig. Screening of hFOB transfected with the pSpCas9(BB)-2A-GFP and pSpCas9(BB)-2A-GFP-gRNA-2 vectors.** (A) Screening by GFP fluorescent microscopy analysis. Representative image of hFOB transfected with the pSpCas9(BB)-2A-GFP vector. Bright field (upper panel) and fluorescent (lower panel) images are shown. (B) Screening by T7 endonuclease assay. Representative gels indicating, in cells targeted with the guide gRNA-2, the presence of the amplicon fragments (arrow) after T7 endonuclease digestion. About 60–70% transfection efficiency and target specificity were demonstrated. MW: molecular weight; gv: digested amplicon from cells transfected with gRNA containing construct; nt: digested amplicon from not transfected cells; nd: not digested amplicon.
(DOCX)

**S2 Fig. Mutant clonal lines screening by T7 endonuclease I assay and restriction enzyme digestions.** (A) The sequences of the gRNAs are indicated on top of the gels, the PAM sequences are in bold. Representative gels indicating the presence of the amplicon fragments (arrow) after T7 endonuclease digestion in cells targeted with gRNA-2 and gRNA-3.2. MW: molecular weight; nd: not digested amplicon, nt: digested amplicon from not transfected cells, ev: digested amplicon from cells transfected with empty vector; gv: digested amplicon from cells transfected with gRNA containing construct. (B) Scheme of the specific restriction enzymes used to discriminate among targeted and WT clones. Two restriction endonucleases recognizing the WT sequence in the region of the Cas9 cleavage were chosen for each guide to optimize the detection of the mutations inserted by non-homologous-end-join repair system. BtsXI and XcmI digestions were performed for gRNA-2 (in orange), BspHI and BccI digestion for gRNA-3.2 (in green). (C) Clonal lines screening for gRNA-2 targeting. Representative gels showing the bands after BstXI and XcmI digestions of the amplicon obtained from gRNA-2 transfected clones. Arrows indicate the bands upon enzymes cleavage. MW: molecular weight; nd: not digested amplicon. (D) Clonal lines screening for gRNA-3.2 targeting. Representative

gels showing the bands after BspHI and BccI digestions of amplicon obtained from gRNA-3.2 transfected clones. Arrows indicate the bands upon enzymes cleavage. MW: molecular weight; nd: not digested amplicon.
(DOCX)

**S3 Fig. Original uncropped scans of the western blots shown in Fig *1B* and Fig *3C*.** (A) Western blot of TRIC-B and β-Actin of the generated clones. (B) Western blot of ITPR1, 2, 3 isoforms on A3 and control clones with relative total protein staining. (C) Western blot of Collagen type I on A3 and control clones with relative total protein staining. (D) Western blot of Osterix on A3 and control clones with relative total protein staining.
(DOCX)

**S4 Fig. $Ca^{2+}$ mobilization from ER.** (A, B, C) Endogenous $Ca^{2+}$ release induced by the $IP_3$-producing autacoid UTP (100 μM) was abrogated by depletion of the ER $Ca^{2+}$ store with cyclopiazonic acid (CPA; 20 μM).
(DOCX)

**S5 Fig. Validation of further *TMEM38B* KO hFOBs.** (A) Expression analysis of the osteoblastogenic marker *RUNX2*. *RUNX2* expression was reduced in B17 mutant clone with respect to control cells at all time points analysed during hFOB differentiation. (B) Osteoblasts' proliferation analyses. Proliferation was reduced in the majority of the mutant clones analysed. (C) Alkaline phosphatase (ALP) activity analysis. The activity of ALP was significantly reduced in B17, B11 and B14 mutant clones compared to WT at 4, 8, 12 and 15 days of differentiation. (D) Mineralization level by ARS staining. The mineral amount was significantly reduced in the KO clone A12 with respect to WT in both differentiated and not differentiated conditions. $^*p<0.05$.
(DOCX)

**S6 Fig. Expression of osteoblast markers and osteoblast proliferation upon inhibition of $Ca^{2+}$ flux by 2-APB.** (A) Expression of the osteoblast markers *RUNX2*, *SP7* and *COL1A1* was reduced in control cells in which $Ca^{2+}$ flux was inhibited by 2-APB. (B) The incubation of control cells with 2-APB reduced osteoblasts proliferation. Data support the calcium dependence of osteoblast differentiation and proliferation.
(DOCX)

**S1 Table. Enzyme digestion for clones transfected with gRNA2.**
(DOCX)

**S2 Table. Enzyme digestion for clones transfected with gRNA3.2.**
(DOCX)

**S3 Table. Enzyme digestion for clones transfected with both gRNA2 and gRNA3.2.**
(DOCX)

**S1 Raw images.**
(PDF)

## Acknowledgments

We thank Dr Silvia Cotti for initial construct preparation and clonal line screening.

## Author Contributions

**Conceptualization:** Antonella Forlino.

**Data curation:** Laura Leoni, Francesca Tonelli, Roberta Besio, Roberta Gioia, Francesco Moccia, Antonella Forlino.

**Formal analysis:** Laura Leoni, Francesca Tonelli, Roberta Besio, Francesco Moccia, Antonio Rossi, Antonella Forlino.

**Funding acquisition:** Antonella Forlino.

**Methodology:** Laura Leoni, Francesca Tonelli, Roberta Besio, Roberta Gioia, Francesco Moccia, Antonella Forlino.

**Supervision:** Antonella Forlino.

**Validation:** Laura Leoni, Francesca Tonelli, Roberta Besio, Antonella Forlino.

**Writing – original draft:** Laura Leoni, Francesca Tonelli, Roberta Besio, Francesco Moccia, Antonella Forlino.

**Writing – review & editing:** Laura Leoni, Francesca Tonelli, Roberta Besio, Francesco Moccia, Antonio Rossi, Antonella Forlino.

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
