## [Decision Letter · Decision Letter 0]

30 Jun 2021

PONE-D-21-18710

Knocking out TMEM38B in human foetal osteoblasts hFOB 1.19 by CRISPR/Cas9: a model for recessive OI type XIV

PLOS ONE

Dear Dr. Forlino,

Thank you for submitting your manuscript to PLOS ONE. After careful consideration, we feel that it has merit but does not fully meet PLOS ONE’s publication criteria as it currently stands. Therefore, we invite you to submit a revised version of the manuscript that addresses the points raised during the review process.

This article is of interest pending some points that need to be clarified and some others needing manuscript alterations. Following point to point the criticisms raised by the two referees the Authors must answer to each of their comments and amend the paper consequently. Would you please send the revised paper highlighting in red all the alterations made.

We look forward to receiving your revised manuscript.

Kind regards,

Gianpaolo Papaccio, M.D., Ph.D.

Academic Editor

PLOS ONE

Journal Requirements:

These requirements apply both to the main figures and to cropped blot/gel images included in Supporting Information. As such, we ask you to provide images of the full, uncut blots (not the cut strips) as a supplemental file.

Reviewers' comments:

Reviewer's Responses to Questions

**Comments to the Author**

1. Is the manuscript technically sound, and do the data support the conclusions?

Reviewer #1: Yes

Reviewer #2: Partly

2. Has the statistical analysis been performed appropriately and rigorously? 

Reviewer #1: Yes

Reviewer #2: I Don't Know

3. Have the authors made all data underlying the findings in their manuscript fully available?

Reviewer #1: Yes

Reviewer #2: Yes

4. Is the manuscript presented in an intelligible fashion and written in standard English?

Reviewer #1: No

Reviewer #2: Yes

5. Review Comments to the Author

Reviewer #1: The Authors used CRISPR/Cas9 technology to generate an in vitro osteoblast model for OI type XIV, reproducing the molecular and biochemical features of patients’ cells and demonstrating the negative effect of TRIC-B loss-of-function in osteoblast proliferation and mineralization.

The manuscript is very interesting. Although this, there are some points that need to be addressed.

The authors should perform also TRIC-B Known down with siRNA and compare the results between two methodologies (CRISPR/Cas9 and siRNA). The Authors performed a serial dilution to select clones, but they did not indicate the percentage of transfection. To confirm their model, the authors must evaluate ALPL, OPN, IBSP, and DMP, late markers of osteogenesis that should increase in osteoblasts of OI type XIV (lin Endocrinol Metab. 2017 Jun 1;102(6):2019-2028.). Moreover, the molecular expression of bone related –markers must be confirmed by corresponding protein levels. Then, ALP and Alizarin Red S were evaluated at different times. The assays must be performed at the same times (4, 8 and 15 days). Images of ALP and Alizarin Red S must be added. The subtitle “Cell mineralization” must be changed with: “Extracellular matrix mineralization”. RANKL/OPG profile must be investigated. Discussion section is too long.

Reviewer #2: In this manuscript the Authors attempt to create a cellular model of Osteogenesis imperfecta type XIV by knocking down TRIC-B, a cationic channel specific for potassium and ubiquitously present in the endoplasmic reticulum (ER) membrane. To do that they applied CRISPR/Cas9 technology against TMEM38B, the gene coding for TRIC-B, in human fetal osteoblasts. The authors were able to obtain several clones were TRIC-B was knocked down and selected one clone as the one that recapitulate the disease. The work is interesting in principle, however there few concerns that need to be discussed as follows:

1. CRISPR/Cas9 technology is now quite well known and used, so I’0m not sure there a point in showing all the passages of the protocol in figure 1A.

2. It is not clear why among several clones in which TRIC-B is knocked down, the authors picked the A3, if their hypothesis work (TRIC-B impairment causes OI type XIV) all the clones (or at least few of them) must recapitulate the disease characteristics. How can we be sure that the clone A3 is not obtained just by chance? This protocol should reproducible in different labs and so the authors should show that they can have more than one clone reproducing the disease.

3. Some markers are analyzed just by real time PCR, these results must be confirmed at protein level.

4. In figure 4 C, the legend is missing and is not clear what they are evaluating. In figure 4D, “differentiation factors” is misleading, that is the absorbance of alizarin red and that should be stated clearly. Images of Alizarin red well must be shown.

5. The discussion is too long, I don’t think there is a point is discussing CRISPR/Cas9 drawback and troubleshooting.

6. The quality of English composing can be improved. The sentences are sometimes too long and difficult to comprehend.

6. PLOS authors have the option to publish the peer review history of their article (what does this mean?). If published, this will include your full peer review and any attached files.

Reviewer #1: No

Reviewer #2: No

---

## [Author Response · Author response to Decision Letter 0]

11 Aug 2021

Dear Professor Gianpaolo Papacci,

Academic Editor of PlosOne

We are submitting the revision of the manuscript “Knocking out TMEM38B in human foetal osteoblasts hFOB 1.19 by CRISPR/Cas9: a model for recessive OI type XIV” Manuscript Number: PONE-D-21-18710 by Leoni L. et al.

A manuscript version with track changes is also provided.

We thank the Reviewers for their constructive comments, which we took into careful consideration and we believe that they contributed to improve the manuscript.

The following is a point-by-point response to the Referees comments.

REVIEWERS' COMMENTS

Reviewer #1: 

-The Authors used CRISPR/Cas9 technology to generate an in vitro osteoblast model for OI type XIV, reproducing the molecular and biochemical features of patients’ cells and demonstrating the negative effect of TRIC-B loss-of-function in osteoblast proliferation and mineralization.

The manuscript is very interesting. Although this, there are some points that need to be addressed.

The authors should perform also TRIC-B Known down with siRNA and compare the results between two methodologies (CRISPR/Cas9 and siRNA). 

We thank the Reviewer for the appreciation of our work. For what concern the TRIC-B knock down, we did not perform a gene silencing by siRNA since its transient nature would not have allowed us to evaluate the long term effects we were interested on, i.e. mineralization and late osteoblast markers expression. Nevertheless, to add further evidences to support our findings, we now provide data on other CRISPR Cas generated clones in Fig S5.

-The Authors performed a serial dilution to select clones, but they did not indicate the percentage of transfection. 

The percentage of transfection was 60-70%, as indicated in Supplementary Materials- results section “using the constructs pSpCas9(BB)-2A-GFP and pSpCas9(BB)-2A-GFP-gRNA-2, a 60-70% transfection efficiency and target specificity were demonstrated (Supplementary Figure S1A, B)”. We did not perform such evaluation on the cells transfected using pSpCas9(BB)-2A-PURO since we expected similar efficiency and we did not have the fluorescence marker to use for proper quantitation.

-To confirm their model, the authors must evaluate ALPL, OPN, IBSP, and DMP, late markers of osteogenesis that should increase in osteoblasts of OI type XIV (lin Endocrinol Metab. 2017 Jun 1;102(6):2019-2028.). Moreover, the molecular expression of bone related –markers must be confirmed by corresponding protein levels. 

As requested, we performed the analyses of the molecular expression of the suggested late osteoblastogenesis markers. ALPL expression was reduced in A3 compared to controls, in agreement with the ALP activity data. IBSP, and DMP were increased, as in the human cells. OPN was instead decreased, in agreement with the generally opposite roles of IBSP and OPN in promoting and opposing mineralization (Bouleftour W, et al. The role of the SIBLING, bone sialoprotein in skeletal biology—contribution of mouse experimental genetics. Matrix Biol. 2016;52-54:60–77). The osterix and collagen I expression profile was confirmed by western blot data. The new data were added in Figure 3 B, C and in the text.

-Then, ALP and Alizarin Red S were evaluated at different times. The assays must be performed at the same times (4, 8 and 15 days). Images of ALP and Alizarin Red S must be added. 

As requested, we performed the ALP activity measurement also at day 15 and the new data are shown in Figure 4C. We also added the images of the alizarin red S mineral staining in Figure 4D.

-The subtitle “Cell mineralization” must be changed with: “Extracellular matrix mineralization”. 

We revised the subtitle following the suggestion.

-RANKL/OPG profile must be investigated. 

As requested, the RANKL/OPG profile was investigated by qPCR. The ratio was increased in the mutant clone at 4 and 21 days of culture, in agreement with Webb et al. data (Endocrinol Metab. 2017 Jun 1;102(6):2019-2028), suggesting an effect on osteoclastogenesis. We added the new data in Figure 3D and in the text.

-Discussion section is too long.

We agree with the Reviewer and we shortened the discussion. In particular, we removed the section focusing on Crispr/Cas9 limitation. We believe it will not compromise the message of the manuscript and it will make easier reading the discussion.

Reviewer #2:

In this manuscript the Authors attempt to create a cellular model of Osteogenesis imperfecta type XIV by knocking down TRIC-B, a cationic channel specific for potassium and ubiquitously present in the endoplasmic reticulum (ER) membrane. To do that they applied CRISPR/Cas9 technology against TMEM38B, the gene coding for TRIC-B, in human fetal osteoblasts. The authors were able to obtain several clones were TRIC-B was knocked down and selected one clone as the one that recapitulate the disease. The work is interesting in principle, however there few concerns that need to be discussed as follows:

1. CRISPR/Cas9 technology is now quite well known and used, so I’m not sure there a point in showing all the passages of the protocol in figure 1A.

We agree the CRISPR/Cas9 genome editing is indeed a relatively common and well-known technique and indeed most of the information on mutant generation has been added as supplementary materials, nevertheless, to better follow the flow of our experiments, we believe it is useful to keep the scheme in the main text (Fig. 1A).

2. It is not clear why among several clones in which TRIC-B is knocked down, the authors picked the A3, if their hypothesis work (TRIC-B impairment causes OI type XIV) all the clones (or at least few of them) must recapitulate the disease characteristics. How can we be sure that the clone A3 is not obtained just by chance? This protocol should reproducible in different labs and so the authors should show that they can have more than one clone reproducing the disease.

We agree with the Reviewer, in order to proof the goodness of the model, more than one clone should recapitulate the human disease characteristic. Indeed, we obtained several knocked out clones for TRIC-B but, we decided starting from Figure 3 to show data only for clone A3 to avoid redundancy and large images. Nevertheless, following the Reviewer’ recommendation, we now provided further validation of our findings on different TRIC-B knock out clones in Fig S5.

3. Some markers are analyzed just by real time PCR, these results must be confirmed at protein level.

As requested also by Reviewer 1, we confirmed at the protein level the expression of some osteoblastogenic markers (osterix and collagen type I). We added the new data in Figure 3C and in the text.

4. In figure 4 C, the legend is missing and is not clear what they are evaluating. In figure 4D, “differentiation factors” is misleading, that is the absorbance of alizarin red and that should be stated clearly. Images of Alizarin red well must be shown.

We apologize for the missing/misleading legend in Figure 4 C and D. We fixed it. As requested, we added images of the alizarin red staining in Figure 4D.

5. The discussion is too long, I don’t think there is a point is discussing CRISPR/Cas9 drawback and troubleshooting.

We agree with the Reviewer and we shortened the discussion removing the CRISPR/Cas9 drawback and troubleshooting section. 

6. The quality of English composing can be improved. The sentences are sometimes too long and difficult to comprehend.

As requested, the English has been revised.

---

## [Decision Letter · Decision Letter 1]

27 Aug 2021

Knocking out TMEM38B in human foetal osteoblasts hFOB 1.19 by CRISPR/Cas9: a model for recessive OI type XIV

PONE-D-21-18710R1

Dear Dr. Forlino,

We’re pleased to inform you that your manuscript has been judged scientifically suitable for publication and will be formally accepted for publication once it meets all outstanding technical requirements.

Kind regards,

Gianpaolo Papaccio, M.D., Ph.D.

Academic Editor

PLOS ONE

Additional Editor Comments (optional):

Reviewers' comments:

Reviewer's Responses to Questions

**Comments to the Author**

1. If the authors have adequately addressed your comments raised in a previous round of review and you feel that this manuscript is now acceptable for publication, you may indicate that here to bypass the “Comments to the Author” section, enter your conflict of interest statement in the “Confidential to Editor” section, and submit your "Accept" recommendation.

Reviewer #1: All comments have been addressed

Reviewer #2: All comments have been addressed

2. Is the manuscript technically sound, and do the data support the conclusions?

Reviewer #1: Yes

Reviewer #2: Yes

3. Has the statistical analysis been performed appropriately and rigorously? 

Reviewer #1: Yes

Reviewer #2: I Don't Know

4. Have the authors made all data underlying the findings in their manuscript fully available?

Reviewer #1: Yes

Reviewer #2: Yes

5. Is the manuscript presented in an intelligible fashion and written in standard English?

Reviewer #1: Yes

Reviewer #2: Yes

6. Review Comments to the Author

Reviewer #1: (No Response)

Reviewer #2: The Authorsa hve addressed all the concern raised by this reviewer. The manuscript is now greatly improved.

7. PLOS authors have the option to publish the peer review history of their article (what does this mean?). If published, this will include your full peer review and any attached files.

Reviewer #1: No

Reviewer #2: No

---

## [Editor Report · Acceptance letter]

17 Sep 2021

PONE-D-21-18710R1 

Knocking out *TMEM38B* in human foetal osteoblasts hFOB 1.19 by CRISPR/Cas9: a model for recessive OI type XIV 

Dear Dr. Forlino:

I'm pleased to inform you that your manuscript has been deemed suitable for publication in PLOS ONE. Congratulations! Your manuscript is now with our production department. 

Kind regards, 

on behalf of

Prof. Gianpaolo Papaccio 

Academic Editor

PLOS ONE